# Muscle and Bone Health in Young Chilean Adults with Phenylketonuria and Different Degrees of Compliance with the Phenylalanine Restricted Diet

**DOI:** 10.3390/nu15132939

**Published:** 2023-06-28

**Authors:** Eugenia Rojas-Agurto, María Jesús Leal-Witt, Carolina Arias, Juan Francisco Cabello, Daniel Bunout, Verónica Cornejo

**Affiliations:** Instituto de Nutrición y Tecnología de Alimentos INTA, Universidad de Chile, Santiago 7830490, Chile; eugenia.rojas@inta.uchile.cl (E.R.-A.); carias@inta.uchile.cl (C.A.); jfcabello@inta.uchile.cl (J.F.C.); dbunout@inta.uchile.cl (D.B.); vcornejo@inta.uchile.cl (V.C.)

**Keywords:** inborn error of metabolism, phenylalanine, muscle mass, muscle function

## Abstract

There are concerns about muscle and bone health in patients with Phenylketonuria (PKU). Our aim was to compare muscle mass, function, and bone health among young adults with PKU who maintained or suspended dietary treatment. Methods: Three groups were considered—PKU-1: 10 patients who used a protein substitute (PS) without phenylalanine (Phe); PKU-2: 14 patients who used the PS without Phe until eighteen years old and then practiced mostly a vegan diet; and 24 matched healthy controls. A 24 h recall survey, blood parameters, body composition and bone mineral density through DEXA, rectus femoris thickness by ultrasound, hand grip strength, submaximal exercise test, and walking speed were assessed. Results: PKU-1 patients had lower hand grip strength than their matched controls, but no other differences. Compared to controls, the PKU-2 group had lower fat-free mass (*p* = 0.01), less spine and femoral bone mineral density (*p* = 0.04 and *p* < 0.01, respectively), and peak workload on the incremental test (*p* = 0.03). When comparing PKU groups, blood Phe levels were significantly lower in the PKU-1 group (*p* = 0.02). Conclusions: Among PKU patients, abandoning the dietary treatment and maintaining high blood Phe concentrations could be deleterious for muscles and bones. However, we cannot discard other causes of bone and muscle damage in these patients.

## 1. Introduction

Phenylketonuria (PKU, OMIM 261600) is the consequence of an autosomal recessive phenylalanine hydroxylase deficiency and requires early diagnosis and the dietary restriction of phenylalanine (Phe) in the neonatal period to avoid devastating consequences on cognitive performance. A high concentration of Phe in adults affects executive function and working memory, but the effects that it can produce in the rest of the organism are still unknown [1,2,3]. Early diagnosis and timely treatment reverse neurological disorders, but to achieve this goal, dietary management must consider some aspects. Specifically, a reduction of Phe intake requires a restriction of natural protein intake, and proteins that provide less Phe, to prevent excess Phe concentration in blood and brain, should be selected. In addition, low blood Phe levels must be maintained for life [4,5]. The Chilean protocol indicates that blood Phe levels between 120–360 μmol/L represent good metabolic control, similar to U.S. guidelines [6,7]. In the case of European protocol, subjects over 12 years old were allowed a Phe concentration of up to 600 μmol/L [4]. Dietary treatment prohibits the intake of foods such as meat, dairy products, fish, shellfish, eggs, and legumes, due to their high Phe content, and restricts the intake of cereals, fruits, and vegetables. As a result of the restriction of natural foods, various nutrients are not adequately provided by the diet, making it necessary to deliver a protein substitute without Phe (PS) to replace the natural proteins that are not allowed in the diet. This PS is an amino acid mixture supplemented with tyrosine, minerals, and vitamins to avoid micronutrient deficiencies [6]. The tolerance to the amount of Phe provided by the diet will depend on the genotype variant of the Phe hydroxylase enzyme involved, where some patients tolerate a lower amount of natural proteins to maintain an optimal Phe concentration and others tolerate more quantity of Phe [4,7].

Since 1992, Chile has had a national neonatal screening program for PKU, and the state subsidizes the PS without Phe for all PKU children. However, the program originally subsidized the formula only until 18 years of age [8], which implied that many patients who had maintained an adequate nutritional treatment until the age of 18, opted for a vegan diet since they could not afford the cost of PS without Phe. This omission was corrected in 2017, when the state expanded the PS subsidy for all patients, regardless of age. Thus, two groups of PKU patients were indirectly created: those who remained on nutritional treatment and metabolic control and those who stopped taking the PS and no longer had metabolic controls but maintained a mostly vegan diet.

There are concerns about the effects that the restriction of intact proteins may have on body composition and muscle function in PKU patients. Previous studies show discordant results, some reporting an increase in fat mass [9] and a reduction in muscle mass [10] and others not showing differences with age and gender-matched controls [11]. Some reports also suggested a direct toxic effect of high Phe concentration on muscle [10]. A recent review concluded that there are inconsistent data about the effects of protein restriction or high Phe levels on muscle mass and function in patients with PKU. Noteworthy is that only seven studies used dual-energy X-ray absorptiometry to assess muscle mass [12].

Two previous studies reported a normal or lower peak oxygen consumption (VO_2_max) in these patients during exercise tests [13,14]. A lower muscle mass should result in lower muscle strength and performance in functional tests such as walking capacity or incremental cardiopulmonary tests [15]. The effects of the disease on muscle mass could lead to a higher cardiovascular risk or functional impairment during adulthood. It is known that functional capacity at the early stages of life is associated with a higher risk of functional impairment later in life [16].

Bone health is also a concern. Lower bone mineral density in patients with PKU was reported in a meta-analysis of 13 studies assessing bone mineral density [17]. Hypotheses suggest an effect of dietary restriction or a higher urinary calcium excretion due to the high acid load delivered by the PS without Phe, an increased kidney workload, and a higher inflammatory response [18].

We studied muscle mass and function and bone mineral density in two groups of young adults with PKU, one that was on nutritional treatment, received the PS, and attended metabolic controls, and another group that had stopped taking PS had not attended metabolic controls since the state stopped subsidizing the PS at the age of 18, and maintained a mostly vegan diet. We compared the results obtained in these two PKU groups with healthy controls matched on age, sex, and body mass index. Our aim was to assess if these two groups had any differences in body composition or bone mineral density or if they differed from matched controls.

## 2. Materials and Methods

Between August 2019 and January 2020, the database of the Diagnostic Center belonging to the Institute of Nutrition and Food Technology (INTA) of the University of Chile was reviewed to identify potential participants. A total of 24 patients with PKU who began nutritional treatment from neonatal diagnosis were identified. Two groups of patients with PKU older than 18 years old with the following characteristics were studied:

PKU-1: 10 Participants with a neonatal diagnosis of PKU, who continued with nutritional treatment, received an adequate supply of PS without Phe, and kept strict follow-up.

PKU-2: 14 participants with a neonatal diagnosis of PKU, discontinued the PS at 18 years of age (between one and seven years before this assessment), because they did not receive a state subsidy, and stopped attending metabolic control appointments. Although, they stopped the critical micronutrient supplementation too (calcium, iron, and zinc).

Before the start of the study, each group was characterized with respect to its participants’ adherence to treatment in the two preceding years. To be classified as adherent to treatment, we considered data corresponding to metabolic control (average phenylalanine concentration measured in dried blood spots), attendance at follow-up appointments, and regular intake of protein substitutes without phenylalanine.

Matched control group: A total of 24 controls were recruited and matched by age, sex, and body mass index (BMI; kg/m^2^) with PKU-1 and PKU-2 patients. We excluded pregnant women, subjects with another metabolic disease, subjects with a moderate or profound intellectual and physical disability, and those who refused to sign an informed consent.

All participants were assessed at INTA, University of Chile, where they were invited to attend after an overnight fast (between eight to 12 h). The following assessments were carried out:

### 2.1. Anthropometry

We measured weight (kg) and height (m) with a Seca scale (0.005 kg accuracy) and a stadiometer (0.01 cm margin of error), respectively. Waist circumference (WC) was measured at the middle point between the last rib and iliac crest and expressed in cm. BMI was calculated.

### 2.2. Analytical Methods

Phe and Tyrosine (Tyr) concentrations were measured in a sample obtained from dried blood on a filter paper card by tandem mass spectrometry at INTA, which is accredited by the superintendence of health (Ministry of Health of Chile). Values between 120 and 360 μmol/L for Phe and between 44 and 99 μmol/L for Tyr were considered a good metabolic control for the adult PKU population.

A fasting blood sample was obtained to measure folic acid, serum lipids, vitamins B12 and D [19], and homocysteine at a certified clinical laboratory.

### 2.3. Body Composition and Bone Mineral Density

Body composition and bone mineral density were measured by dual-energy X-ray absorptiometry using GE Lunar iDEXA equipment (GE Healthcare, United States). Participants had a normal fluid intake the day prior to the evaluation. Normal hydration status was confirmed clinically, looking for the absence of dehydration signs such as peripheral edema or orthostatic hypotension. General Electric software version 13.6 was used. The measurement error of the method for body composition is 2.9% [20]. Both spine and femoral bone mineral densities were also measured. Z scores for spine and femoral neck bone mineral density were calculated using data from NHANES 2009–2010 examination (https://wwwn.cdc.gov/nchs/nhanes/search/default.aspx) (accessed on 25 May 2023), according to age and sex.

The participants received a standard breakfast (using Phe free products for PKU patients) and thereafter were subjected to the following assessments:

### 2.4. Rectus Femoris Ultrasound

Rectus femoris ultrasound using a General Electric LogiQ ultrasound device (GE Healthcare, United States). With the participant in the supine position and the legs relaxed, the probe was placed perpendicular to the skin without exerting compression, over the mid portion of the rectus femoris, calculated as half the distance between the anterior superior iliac spine and the lower edge of the patella. This landmark was chosen because it is easy to determine and reproducible between observers and is usually employed in other studies [21]. Special care was taken to avoid any voluntary contraction of the rectus femoris. A GE 12L-RS 14.2 × 47 mm linear array ultrasound transducer probe was used, it was set to measure at a frequency of 8 MHz, with a gain of 58 dB and a depth of 5 cm. Left and right muscles were measured. The cross-sectional thickness of the muscle was measured during the examination and the images were stored. All images were obtained by the same operator. Every measurement was made in triplicate and mean values are reported. The reliability of the method has been previously reported [22].

### 2.5. Physical Performance

Handgrip strength was measured using a Therapeutic Instruments (Clifton, NJ, USA) dynamometer. Three measurements were made in each hand and the higher value obtained for either hand was recorded. Measurement error for repeated assessments in the same individual was 8.7% [23].

Submaximal oxygen consumption during a cardiopulmonary incremental test in a braked cycle ergometer was measured using a Sensormedics Vmax Encore 29 equipment (Sensormedics, Italy). The incremental exercise test was started at a 10-watt ramp with 10-watt increases per minute or a 15-watt ramp with 15-watt increases (depending on the self-reported physical condition of participants) until volitional exhaustion, with a cadence of 60 rpm. The test was stopped if the participant could not maintain the cadence after the stage change or if the respiratory exchange ratio exceeded 1.3 [24]. A face mask was used to collect respiratory gases and the breath-by-breath method was used to measure oxygen consumption and CO_2_ production. Peak oxygen consumption was determined at the moment in which the participant was not able to maintain the cycling cadence. To test the reproducibility of the measurement, we repeated the test within one month in 20 healthy participants. The rho for concordance of two peak oxygen consumption measurements carried out in the same individual was 0.8. The median (interquartile range) peak oxygen consumption values in the first and second measurements were 20.7 (16.1, 27.3) and 21.5 (16.3, 25.3) mL/min/kg [25].

Walking capacity was measured with the six minutes’ walk test (6 MWT) in which the distance covered on a flat surface at a constant pace over six minutes was calculated [26].

### 2.6. Qualitative Measurements

Assessment of regular physical activity: The International Physical Activity Questionnaire (IPAQ) was completed to assess usual physical activity [27]. The activity was expressed as Mets/week [28].

Diet Analysis: Two 24 h (24 h) dietary recalls, one for a weekday and one for a weekend were carried out and both results were averaged to calculate intake. Food portions were determined using a Chilean photographic atlas for typical local food preparations [29]. Macro and micronutrient intake were quantified with the amino acid analyzer (AAA) software created at INTA, version 3.1, with the tables of the chemical composition of Chilean foods [30]. Energy (kcal/d), protein (g/d), cholesterol (mg/d), vitamin B12 (μg/d), folic acid (μg/d), vitamin D (μg/d), calcium (mg/d), Phe (mg/d), and Tyr (mg/d) intakes were estimated. Protein intake was classified as natural protein and protein from PS.

### 2.7. Statistical Analysis

All statistical analyses were performed in Stata 15 for Windows (StataCorp 4905 Lakeway Drive College Station, Texas 77845 USA). As most variables had a non-normal distribution, results are expressed as median (25th–75th centile). Differences between PKU patients and matched controls were assessed with a paired analysis using the Wilcoxon test. Differences between PKU-1 and PKU-2 patients were assessed using Mann-Whitney two-sample statistics and differences in proportions were assessed using the Fisher test. Statistical significance was established at *p* < 0.05.

Ethics committee: The informed consent was approved by the INTA ethics committee in June 2019 (Pt14-2019).

## 3. Results

The demographic, anthropometric features, physical activity questionnaire, results of DEXA, and incremental exercise test of each group of participants are shown in Table 1.

The PKU-1 patients had lower hand grip strength than matched controls. PKU-2 patients had a lower weight and height but similar BMI than controls. They also had a lower peak workload on the incremental cardiopulmonary test, lower fat-free mass, expressed as absolute values or as the appendicular fat-free mass index, and lower spine and femoral bone mineral density.

Laboratory results are shown in Table 2. PKU patients had higher Phe concentration than controls. Additionally, the PKU-1 patients had lower concentrations than their PKU-2 counterparts. Serum folic acid was higher in both groups of PKU patients compared with controls. Serum vitamin D was significantly lower in PKU-2 than in PKU-1 patients. Vitamin B12 levels were also significantly lower in PKU-2 than in PKU-1 groups. No differences in homocysteine levels were observed between groups.

The results of the 24 h dietary recalls are shown in Table 3. Three patients in the PKU-2 group admitted occasional transgressions in the diet and ate meat products (chicken, tuna, egg), included in one of two record days. The remaining eleven maintained a vegan diet without transgressions. PKU-1 patients had lower cholesterol intake than their matched controls. They also had higher intakes of some micronutrients: vitamin B12 (*p* < 0.01), folic acid (*p* < 0.01), vitamin D (*p* < 0.01), and calcium (*p* = 0.04), significantly different too. Furthermore, comparing PKU-1 patients with PKU-2 patients, these subjects had a higher cholesterol intake (*p* = 0.02) and lower calcium intake (*p* = 0.04). Less than 15% of total PKU consumed a low protein food, for this reason, we did not consider this variable in the analysis.

Summarizing the results of Table 1, Table 2 and Table 3, PKU-1 and PKU-2 patients had similar BMI, body composition, and bone mineral density. PKU 2 patients had higher Phe blood levels and lower vitamin B 12 and D levels than PKU-1 patients. As expected PKU-2 patients had lower cholesterol tyrosine, vitamin B 12, vitamin D, and calcium intake than PKU-1 patients.

## 4. Discussion

We observed that PKU-2 patients, who had abandoned the dietary treatment, had lower fat-free mass, bone mineral density, and worse performance on an incremental exercise test compared with matched controls. Controls were healthy participants carefully matched by age, sex, and BMI. This design allowed us to perform a paired comparison of the studied variables, avoiding the distortion caused by sex differences in body composition, bone density, and muscle performance [34].

The lower muscle health markers in the PKU-2 group may indicate that the deleterious effects of the disease derive from the accumulation of Phe possibly generated by the maintenance of dietary restrictions, specifically protein, without professional supervision, due to discontinuation of treatment. However, other factors explain the lower muscle and bone parameters in PKU-2 patients such as protein quality intake, lower vitamin and mineral intake, and lower physical activity derived from the lack of motivation to have healthy lifestyles after having to abandon the dietary treatment. We did not measure prealbumin as an inflammatory marker influencing muscle function. Therefore, we cannot discard its pathogenic importance.

Within the nutritional treatment protocol, PS without Phe is essential to complement the restriction of intact proteins. According to the data extracted from the dietary survey, more than 80% of PKU-1 patients met 100% of their Recommended Dietary Allowance (RDA) of protein with PS and natural proteins (RDA), unlike the PKU-2 group, of whom only 50% met their requirements. A similar study in which PKU patients with normal or elevated Phe concentration were compared, also showed the same pattern of alterations that we observed, namely a lower bone mineral density and lean body mass among patients with higher Phe concentration [35]. The review by Firman et al., also concluded that PKU adversely influences muscle mass [12]. The author also concludes that new studies are required to confirm or refute the pathogenic importance of a lower protein intake on muscle function in PKU patients. We observed that PKU-1 patients had lower handgrip strength as compared with their matched controls. We do not have a plausible explanation for this observation. On the other hand, PKU-2 patients showed a lower peak workload during the incremental test compared with matched controls. These lower muscle function tests showed a trend, suggesting that PKU may be a risk factor for the development of sarcopenia and its functional consequences at younger ages than the general population, reinforcing the need to maintain treatment for life. We have shown previously the association between a lower peak workload with sarcopenia [36]. Physical activity levels were similar between the two groups of patients and controls, indicating that this activity did not have a confounding effect on muscle mass and function. The pathogenic role of low natural protein intake on muscle loss remains to be elucidated. In this study, the vast majority of PKU-2 patients (all but three), despite treatment suspension, maintained a vegan diet, and just three subjects incurred dietary transgressions by consuming proteins of high biological value. However, modifications in protein intake, within 0.8 and 1.3 g/kg/day do not modify muscle mass in patients with sarcopenia [37]. On top of its functional consequences, a lower muscle mass influences cardiovascular health, insulin sensitivity, and thus glycemic status and serum lipid levels.

Although not all reports show a lower muscle mass in PKU patients, the reduction in bone mineral density is a universal finding [38] as shown in a meta-analysis conducted by Demirdas et al. [17]. In our patients, BMD was lower than the normal population reference range, but it was clinically normal. Therefore, our PKU patients were not at a higher risk for fractures. Additionally, experiments in PKU mice suggest that Phe concentration may adversely affect bone metabolism [39], although mice are not a good model to mimic bone formation in humans. However, the meta-analysis did not find an association between intact protein intake (natural protein) or blood Phe concentration and low bone mineral density [17]. Thus, the protein restriction required to manage PKU probably has little or no role in this alteration. Low serum vitamin D is another factor associated with lower bone mineral density in PKU patients [40]. PKU-2 patients had lower vitamin D levels and dietary intake than their PKU-1 counterparts. This is understandable since the PS consumed by the latter is supplemented with the vitamin. The frequency of vitamin D deficiency is high in the general population in Chile and other countries, according to recent surveys [41,42]. In a previous study, we observed that patients with PKU treated with PS had higher vitamin D levels and similar spine and femoral neck bone mineral density than healthy controls, emphasizing the importance of an adequate vitamin D intake to maintain good bone health [43].

The results of the dietary recall of these patients reflect the changes in the diet that are required by the disease. Regarding Phe intake, there were no differences between both PKU groups. However, the high dispersion of intake values and the inherent accuracy of dietary recalls should hinder subtle intake differences between groups. Despite the lack of attendance to routine follow-ups and the lack of PS intake, the PKU-2 patients maintained a predominantly vegan diet, which is low in this amino acid. Just three PKU-2 subjects mentioned eating a little portion of animal-origin food occasionally, but their Phe concentration was not significantly different from the other PKU-2 subjects. Consequently, significant differences in blood Phe levels between PKU-1 and PKU-2 were observed, indicating that the Phe intake tolerance of both PKU groups could be low but the difference response was in the food attitude and transgression in the PKU-2 group. The higher micro-mineral intake in PKU-1 patients is derived from the PS that they use, which is fortified with these nutrients. The low blood vitamin B12 levels in PKU-2 patients once again confirm that despite the discontinuation of treatment they maintained mostly a vegan diet [44]. However, homocysteine levels remained within normal levels despite the lower vitamin B12 levels [45].

The main weakness of this study is the low number of observations. In addition, not measuring prealbumin was a further limitation. However, PKU is a rare disease, and it is difficult to gather a higher number of patients. Furthermore, these patients had a historical Phe value, required to confirm that prior to intervention, PKU-1 and PKU-2 patients had similar blood Phe concentrations (543 μmol/L, 438–730 μmol/L; 617 μmol/L, 429–803 μmol/L, respectively). The main strengths were the possibility to study PKU patients with different levels of compliance to the Phe-restricted diet, to compare them with carefully matched healthy controls, and to incorporate the assessment of muscle function and aerobic capacity.

## 5. Conclusions

We observed a risk for alterations in muscle and bone health in PKU patients, especially those who did not receive the protein substitute, were consuming mostly vegetable proteins, and maintained high Phe concentration. Future research should be addressed to increase the number of subjects and elucidate how high Phe concentration affects muscle performance and bone integrity.

## Figures and Tables

**Table 1 nutrients-15-02939-t001:** Demographic body composition muscle function and bone mineral density data of PKU-1 and PKU-2 groups and controls.

	PKU-1 Patients ^¶^	Matched Controls	*p* ^¥^	PKU-2 Patients *	Matched Controls	*p* ^§^	*p* ^†^
Demographic-anthropometric- physical activity data							
Women/men (*n*)	5//5	5//5		5//9	5//9		NS
Age (years)	23.5 (19–26) ^‡^	21.5 (20–27)	NS	22.5 (18.5–25.5)	23 (19–25)	NS	NS
Weight (kg)	64.6 (58.8–91.4)	68.1 (61.5–79.4)	NS	71.5 (60.4–79.2)	75.1 (61.5–87.6)	0.02	NS
Height (cm)	161 (156–171.2)	166.6 (156–172.4)	NS	162.5 (155.3–168.4)	163.3 (159.5–172.5)	0.02	NS
Body mass index (kg/cm^2^)	24.3 (22.4–28.5)	24.3 (24.1–27.9)	NS	26.7 (24–29.9)	27.6 (23.3–30.6)	NS	NS
Waist circumference (cm)	81.5 (75–94)	79.8 (74–90.2)	NS	85.5 (78.5–102)	83.3 (78.5–105.5)	NS	NS
Physical activity questionnaire (mets/week)	1233 (1040–5124)	1243.8 (942–2862)	NS	1032 (333–4738)	1993.8 (1046–4283)	NS	NS
**Functional measures:**							
Left handgrip strength (kg)	27.2 (21–35.8)	30.9 (24.8–39.7)	0.04	29.7 (22.4–34.7)	32.8 (23.8–39.6)	NS	NS
Right handgrip strength (kg)	30.6 (23.5–40.1)	34 (28.1–42.9)	0.01	34.8 (25.1–37.4)	33.8 (27.6–46.1)	NS	NS
Six minutes’ walk test (m)	650.6 (647–765)	704.8 (674–721.4)	NS	682 (612.7–740.1)	672.5 (638.5–716)	NS	NS
**Muscle ultrasound:**							
Left rectus femoris thickness (mm)	21.2 (20.6–24.2)	22.3 (20.8–25.1)	NS	22.2 (20.5–24.9)	23.5 (20.4–25.7)	NS	NS
Right rectus femoris thickness (mm)	23 (21.4–25.9)	22.7 (21.1–24.8)	NS	22.4 (20.1–23.7)	22.8 (20.9–25.3)	NS	NS
**Incremental exercise test:**							
Peak oxygen consumption (ml/min/kg)	19 (15.9–25.7)	26.2 (21.2–39.1)	NS	23.4 (18.8–31.1)	26.4 (19.5–34.7)	NS	NS
Peak workload (Watt)	135 (103–161)	167.5 (150–233)	NS	144 (116.5–192.5)	167.5 (140.5–218)	0.03	NS
**Double beam X ray absorptiometry (DEXA)**							
Total fat free mass (kg)	44.3 (34.9–51.5)	44.1 (38.6–49.6)	NS	44.4 (36.3–50.5)	47.4 (37.9–57.9)	0.01	NS
Total fat mass (kg)	21.1 (16–30.2)	21.7 (15.4–34.7)	NS	24.4 (18.1–32.4)	26.6 (13.4–36.1)	NS	NS
Appendicular fat free mass (kg)	18.3 (14.3–21.6)	19.1 (15.8–21)	NS	17.7 (14.7–21.7)	19.8 (16.1–25.4)	<0.01	NS
Appendicular fat free mass index (kg/cm^2^)	6.8 (6–7.9)	7.2 (6–7.7)	NS	6.5 (6–7.7)	7.4 (6.4–8.5)	0.02	NS
Spine bone mineral density (g/cm^2^)	1.1 (1.1–1.3)	1.2 (1.1–1.3)	NS	1.2 (1–1.3)	1.3 (1.1–1.4)	0.04	NS
Femoral bone mineral density (g/cm^2^)	1 (0.9–1.2)	1 (1–1.1)	NS	1 (0.9–1.1)	1.2 (1–1.3)	<0.01	NS

^‡^ = median (25th- 75th centile); ^¶^ =PKU-1 patients on dietary treatment with PS and in metabolic control * = PKU-2 patients who ceased dietary treatment with PS and metabolic control; NS = no significant difference. ^¥^ = Significance for differences between PKU-1 patients and their controls. Paired analysis using Wilcoxon test; ^§^ = Significance for differences between PKU-2 patients and their controls. Paired analysis using Wilcoxon test; ^†^= Significance for differences between PKU-1 and PKU-2 patients. Analysis using Mann-Whitney two-sample statistic.

**Table 2 nutrients-15-02939-t002:** Laboratory parameters of participants.

	Reference Values	PKU-1 Patients ^¶^	Matched Controls	*p* ^¥^	PKU-2 Patients *	Matched Controls	*p* ^§^	*p* ^†^
Phenylalanine (μmol/L)	120–360	260.3 (170–642) ^‡^	39.3 (36.3–42.4)	<0.01	781 (636–1035.1)	47.8 (40.6–48.4)	<0.01	<0.01
Tyrosine (μmol/L)	44–99	46.6 (33.1–49.7)	49.7 (44.2–60.7)	NS	35.9 (33.1–55.2)	53 (44.2–60.7)	NS	NS
Total cholesterol (mg/dL)	100–200	139.5 (121–157)	147 (129–174)	NS	133.5 (121–154)	154 (131–185)	NS	NS
HDL cholesterol (mg/dL)	≥40.0	48 (38.4–56.6)	49.4 (48–60.8)	NS	44.9 (38.2–48.9)	46.6 (40.5–50.6)	NS	NS
Triglycerides (mg/dL)	30–150	81.5 (58–104)	100 (68–127)	NS	93.5 (60–146)	95 (82–138)	NS	NS
Serum folic acid (ng/mL)	4.4–31	26.1 (20–30.4)	15.2 (12.5–19.7)	<0.01	22.4 (18.5–28.1)	16.8 (12.9–18.4)	<0.01	NS
Vitamin B12 (ng/mL)	197–771	722.5 (499–926)	374.5 (291–696)	NS	338 (137–539)	436.5 (330–543)	NS	0.03
Homocysteine (μmol/L)	4.30–11.10	5.1 (4.3–5.9)	6.2 (5.3–6.4)	NS	5.6 (4.9–6.4)	6 (5.2–6.6)	NS	NS
Vitamin D3 (pg/mL)	≥30.0	35.7 (28.4–46.6)	23.7 (23–37.5)	NS	23.9 (19–26.8)	29 (24.8 - 32.3)	NS	<0.01

^‡^ = median (25th- 75th centile); ^¶^ = PKU-1 patients on dietary treatment with PS and in metabolic control; * = PKU-2 patients who ceased dietary treatment with PS and metabolic control; NS= no significant difference; ^¥^ = Significance for differences between PKU-1 patients and their controls. Paired analysis using Wilcoxon test; ^§^ = Significance for differences between PKU-2 patients and their controls. Paired analysis using Wilcoxon test; ^†^ = Significance for differences between PKU-1 and PKU-2 patients. Analysis using Mann-Whitney two-sample statistic.

**Table 3 nutrients-15-02939-t003:** Macronutrient and micronutrient intake- derived from dietary recalls.

	Reference Values	PKU-1 Patients ^¶^	Matched Controls	*p* ^¥^	PKU-2 Patients *	Matched Controls	*p* ^§^	*p* ^†^
Protein (g) (%MDR)	0.8 g/kg/d ^a^ 10–35% ^b^	75.1 (57.3–78.2) ^‡^ 15% (10–20)	84.4 (57.9–101) 15% (13–16)	NS	46.6 (28.9–68.7) 9% (7–13)	89.6 (66.6–101.1) 15% (12–18)	0.01	NS
Fat (g) (%MDR)	20–35% ^b^	55.8 (38–71) 27% (21–29)	97 (70–121) 37% (32–41)	0.01	67.8 (44.5–84.5) 30% (23–37)	86.5 (66.5–95.5) 34% (29–38)	NS	NS
Carbohydrates (g) (%MDR)	45–65% ^b^	269.3 (198–305) 57 % (52–64)	268.3 (226–335) 48% (44–52)	NS	298.5 (245.5–442.5) 61% (55–67)	310 (188.5–420) 51% (45–59)	NS	NS
Energy (Kcal/d)	2500 ^c^	1800.5 (1565–2167)	2263.3 (1923–2850)	NS	1938.3 (1730.5–3068)	2238 (1754.5–3052)	NS	NS
Cholesterol (mg/d)	<300 ^d^	1.3 (0–10.5)	273.5 (181.5–496)	<0.01	15.8 (10.5–69)	283 (158–620.5)	<0.01	0.02
Phenylalanine (mg/d)	200–1100 ^e^	600 (400–800)	3900 (2600–4900)	<0.01	1200 (500–1700)	4000 (3100–4500)	<0.01	NS
Tyrosine (mg/d)	4000–6000 ^e^	5600 (4400–7000)	3200 (2000–3900)	0.04	2700 (1200–4300)	3300 (2500–3600)	NS	0.01
Vitamin B12 (μg/d)	2.4 ^a^	13.5 (11.2–21.4)	5 (3.8–6.5)	<0.01	2.7 (1.1–11.2)	5.3 (2.4–5.6)	NS	<0.01
Folic acid (μg/d)	400 ^a^	1361 (737.5–1869)	416.8 (228.5–498)	<0.01	1672.8 (1404–2941)	458.5 (225–1114)	<0.01	NS
Vitamin D (μg/d)	5 ^a^	11.7 (7.7–13.9)	1 (0.5–1.7)	<0.01	2.8 (1.9–9.3)	1.3 (0.3- 1.9)	0.03	0.02
Calcium (mg/d)	1000 ^a^	2121 (1482.5–2529)	803 (535.5–1119)	0.04	929.5 (501.5–1431.5)	735.3 (466–1221.5)	NS	0.01

^a^ Recommended Dietary Allowance (RDA) [31]; ^b^ Reference values: Acceptable Macronutrient Distribution Ranges (AMDR) percentage about kilocalories per day [31]; ^c^ FAO/OMS/UNU 2007, considering moderate activity [32]; ^e^ Chilean PKU protocol [6]; ^d^ American Heart Association recommendation [33]; ^‡^ = median (25th–75th centile); ^¶^ = PKU-1 patients on dietary treatment with PS and in metabolic control; * = PKU-2 patients who ceased dietary treatment with PS and metabolic control; NS= no significant difference; ^¥^ = Significance for differences between PKU-1 patients and their controls. Paired analysis using Wilcoxon test; ^§^ = Significance for differences between PKU-2 patients and their controls. Paired analysis using Wilcoxon test; ^†^ = Significance for differences between PKU-1 and PKU-2 patients. Analysis using Mann-Whitney two-sample statistic. %MDR: Macronutrient distribution range percentage.

## Data Availability

The data can be obtained by contacting the first author, M.J.L.-W.

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
