# Peer review of "Muscle and Bone Health in Young Chilean Adults with Phenylketonuria and Different Degrees of Compliance with the Phenylalanine Restricted Diet"

_nutrients, 2023, doi:10.3390/nu15132939_

Round 1

Reviewer 1 Report

Dear authors,

Thank you very much for your interesting investigation.

I just have a few questions that need to be answered.

Methods

-          Could you explain how the vegan diet was performed, since it is not easy to perform a healthy vegan diet? Did the patients take any supplement? Was the protein intake monitored regarding quality and/or quantity?

Table 1

-          Do you have any explanation of the large differences regarding “total fat mass” from PKU patients to controls? Why is the difference not significant? Or are the numbers typos?

Table 3

-          Could you add the reference values for the dietary intake of the mentioned nutrients?

-          Is there a reason for not measuring iron?

-          For PKU-2 patients you mention intake of cholesterol and B12. However, vegan diets do not contain any. Can you explain?

Author Response

We thank the reviewer for all their comments and suggestions. We have edited the manuscript (changes highlighted in yellow in the marked-up version) and provide detailed responses to each point raised below (indicated by Response before each response and blue font).

  1. Could you explain how the vegan diet was performed, since it is not easy to perform a healthy vegan diet? Did the patients take any supplement? Was the protein intake monitored regarding quality and/or quantity?

Response: In this case the vegan diet was eaten by each patient, avoiding in their diets every food of animal origin, considering just vegetables, fruits, and cereals. The macronutrients, come from this kind of foods, and in the case of protein, it had a low biological value. No patients was taking supplements to cover the critical micronutrients at the study moment. We added this clarification on the manuscript.

  1. Do you have any explanation of the large differences regarding “total fat mass” from PKU patients to controls? Why is the difference not significant? Or are the numbers typos?

Response: Effectively this was a typo, which was corrected.

  1. Table 3

-          Could you add the reference values for the dietary intake of the mentioned nutrients?

-          Is there a reason for not measuring iron? 

-          For PKU-2 patients you mention intake of cholesterol and B12. However, vegan diets do not contain any. Can you explain?

Response: We added a column with dietary reference intakes of these nutrients. We did not measure iron since we did not consider a relevant nutrient for bone health. As shown in table 3 vitamin b12 and cholesterol intakes were negligible in PKU 2 patients. We agree that vegan diets should not include these nutrients but, as we explain in the results section “Three patients in the PKU-2 group ate meat products (chicken, tuna, egg)”. These transgressions explain the mean intakes of vitamin B12 and cholesterol.

Reviewer 2 Report

Dear Authors

Thank you for your work. I have made some suggestions and challenges to your paper. Overall it adds to the growing literature on muscle function in the vulnerable group. Please see the reommended changes/ suggestions

Overview

Overall, the paper is interesting and adds to the growing evidence that muscle mass/ strength is an important consideration and one that we need to improve and monitor in this population.  There are corrections/ improvements that could be made to strengthen the work.

·       Compare muscle mass function and bone health in young adults with PKU  some on diet some off diet.

·       3 gps:  10 PKU subjects on diet and PS, 14 on diet until 18y and then vegan diet with no PS, 24 controls two groups for PKU 1 and 2 gps

·       24 h recall survey and blood parameters, body composition and BMD via DXA hand grip, test walking speed

·       PKU1 lower hand grip strength than controls but no other differences, PKU 2 lower fat free mass less spine and BMD and more difficult workload on exercise compared to control

·       Bloods phe significantly lower in PKU1 compared with PKU2

·       Conclusion stopping diet treatment and higher blood phe is deleterious for muscle and bones. I would suggest may not just be due to higher Phe concentrations as the dietary intake of group 2 is very different. Finding the exact cause of why the PKU group 2 had lower FFM etc is important, dietary Phe may be a causative factor but other factors need to be considered. The only difference in PKU1 and controls was hand grip strength, but PKU2 gp had normal hand grip strength.

Introduction

·       Low phe levels is this for all ages. In the European guidelines there are two different Phe concentrations depending on age. It would be good to clarify this age range.

·       Interesting that fruit and vegetables are restricted as these are very low in Phe, would the authors be able to expand slightly. What cut off point is used to allow the use of foods freely, or is all phenylalanine counted in the diet.

·       What about low protein foods?

Methods

·       Magnesium, zinc, were these measured as they are important to bone health. Did any subjects controls or PKU take any supplements for vitamins and minerals.

·       Standardisation of gp 2 out of treatment range 1 to 7y is large time frame.

Results

·       Would be good to have a summary of the results to go along with the table- a brief synopsis helps the reader.

·       Overview: a short overview for  each section e.g. age, weight height gp 1 mean age X, range (X_X), nos males and females etc.  For gp 2 there is a difference in weight and height. This only has to be short but helps.

·       Typo in total fat mass for controls 2.4 ? 24

·       Mean nos of years off dietary PS- it would be useful to know how long the PKU2 gp were off treatment as 1-7 years is very different.

·       Comparison of PKU 1 and PKU 2 groups would be useful for all the results section.

Discussion

·       Would suggest re wording as not adequately compliant as they had no choice but to stop dietary treatment at 18 years.

·       Would give a synopsis of the main findings: PKU 1 gp compared to controls had no significant difference except  for hand grips. PKU2 gp with controls and PKU1 and 2.

·       How do you explain that PKU2 gp did not have lower hand grip- if you are saying Phe is the cause this is not the case for this group?

·       I would challenge your statement that it is only Phe concentrations leading to the differences in the findings there are multiple reasons for lower muscle health.

o   protein quality intake

o   vitamin and mineral intake this was different in the groups

o   higher phe may lead to less physical activity as decreased motivation to do exercise and this may affect muscle strength although this was not suggested in your findings.

o   there were no markers on prealbumin which has been shown to affect muscle strength in PKU subjects and could be cited in your discussion.

o   other areas are BCAA intake which might have been a factor in those on a vegan diet

o   the role of leucine in muscle function?

·       Similar study reference 33  need to expand on this and give a little more background.

·       Not just PKU but need to be specific Firman concluded that regarding body protein status in PKU findings were inconclusive and the relationship between diet and protein status  outcome is unclear. This was a detailed review and explored many factors that could influence protein status.

·       Findings from 2 systematic reviews: BMD is it lower than the normal population reference range but it is clinically normal, and this is  an important finding. PKU patients on a diet have normal bone mineral density and are not at risk of increased fractures. I would quote the second reference on bone density in PKU

·       Mice are not the same as humans – this needs to be put into context and one cannot  compare bone formation in mice and humans. Bone formation is a complex process and one animal model cannot be held up as an example of low bone density in PKU. This study looked at CGMP which is a different protein substitute type and leads to another possibility- again not proven - to benefit bone health.

·       Main findings lower peak workload is an important observation but ideally need more explanation and the link with sarcopenia.

·       Muscle mass is an important element for physical and functional ability and plays an underappreciated metabolic role in reducing risk of cardiometabolic diseases, obesity, cardiovascular disease, diabetes and hyperlipidaemia- these points could be highlighted.

·       Limitations did not measure pre albumin which might give a better insight into protein status.

·       Muscle mass does not directly relate to functional ability which is a further limitation and needs more investigation.

Author Response

Reviewer 2:

  1. Conclusion stopping diet treatment and higher blood phe is deleterious for muscle and bones. I would suggest may not just be due to higher Phe concentrations as the dietary intake of group 2 is very different. Finding the exact cause of why the PKU group 2 had lower FFM etc is important, dietary Phe may be a causative factor but other factors need to be considered. The only difference in PKU1 and controls was hand grip strength, but PKU2 gp had normal hand grip strength.

Response: We agree with the reviewer that the conclusion is too emphatical. We added a sentence to the conclusions stating that “we cannot discard other causes for bone and muscle damage in these patients”.

Introduction:

  1. Low Phe levels is this for all ages. In the European guidelines there are two different Phe concentrations depending on age. It would be good to clarify this age range.

Response: Our management protocol, as the United States protocol, indicates a Phe concentration between 120 and 360 umol/L. However, the European protocol is more flexible in patients older than 12 years old and allows a concentration up to 600 umol/L. We clarify these difference on the background.

  1. Interesting that fruit and vegetables are restricted as these are very low in Phe, would the authors be able to expand slightly. What cut off point is used to allow the use of foods freely, or is all phenylalanine counted in the diet.

Response: There are fruits and vegetables with a higher Phe content. The Phe amount is established according to the guidelines proposed by Singh et al (ref added) and according to patients’ blood Phe concentrations. We added this reference.

Singh RH, Cunningham AC, Mofidi S, Douglas TD, Frazier DM, Hook DG, Jeffers L, McCune H, Moseley KD, Ogata B, Pendyal S, Skrabal J, Splett PL, Stembridge A, Wessel A, Rohr F. Updated, web-based nutrition management guideline for PKU: An evidence and consensus based approach. Mol Genet Metab. 2016 Jun;118(2):72-83.

  1. What about low protein foods?

Response: The availability and used of low protein foods is minimal in our country because are expensive and they are not subsidized by the state. Despite their low availability, we recorded their intake in the dietary surveys and just 3 patients pf PKU-1 and 1 patient of PKU-2 referred the occasional use of low protein foods. We observed less than 15% of total PKU subjects consumed low protein food, and for this reason we didn't consider this variable in the analysis. We added in the result “Less than 15% of total PKU consumed a low protein food, for this reason we didn’t consider this variable in the analysis”, to clarify the topic.

  1. Magnesium, zinc, were these measured as they are important to bone health. Did any subjects controls or PKU take any supplements for vitamins and minerals.

Response: Magnesium and zinc levels were not measured in these patients. We agree with the reviewer that these micronutrients are important to bone health. PKU 1 patients were receiving vitamin and mineral supplements contained in the protein substitute, but PKU 2 patients were not receiving supplements since, as stated in material and methods they “stopped attending metabolic control appointments”.

  1. Standardisation of gp 2 out of treatment range 1 to 7y is large time frame.

Response: We agree with the reviewer that the time lapse is high. However, as from one year after discontinuing the low Phe diet, the psychological executive consequences of the disease start to appear, expressed clearly in: Spronsen, Francjan J. van, Annemiek MJ van Wegberg, Kirsten Ahring, Amaya Bélanger-Quintana, Nenad Blau, Annet M. Bosch, Alberto Burlina, et al. «Key European Guidelines for the Diagnosis and Management of Patients with Phenylketonuria». The Lancet Diabetes & Endocrinology 5, n.o 9 (1 de septiembre de 2017): 743-56. https://doi.org/10.1016/S2213-8587(16)30320-5.

Results:

  1. Would be good to have a summary of the results to go along with the table- a brief synopsis helps the reader.

Response: The summary of the results depicted in table 1 is written in the second paragraph of the results section. Quote “PKU-1 patients had lower hand grip strength than matched controls. PKU-2 patients had a lower weight and height but similar BMI than controls. They also had a lower peak workload on the incremental cardiopulmonary test, lower fat free mass, expressed as absolute values or as the appendicular fat free mass index, and lower spine and femoral bone mineral density”. If we add all the numbers shown in table 1, this section will be notoriously enlarged.

  1. Overview: a short overview for each section e.g. age, weight height gp 1 mean age X, range (X_X), nos males and females etc. For gp 2 there is a difference in weight and height. This only has to be short but helps.

Response: Same explanation as the previous one. We tried to summarize in the text of the results section the numbers shown in tables one, two and three. But showing the same numbers of the tables in text will be a duplication. We hope that the reviewer understands the two previous explanations.

  1. Typo in total fat mass for controls 2.4?

Response: Effectively, this was a typographical error that we corrected.

  1. Mean nos of years of dietary PS- it would be useful to know how long the PKU2 gp were off treatment as 1-7 years is very different.

Response: We added a paragraph in the material and methods section, stating that “Before the start of the study, each group was characterized with respect to its participants’ adherence to treatment in the two preceding years. To be classified as adherent to treatment, we considered data corresponding to metabolic control (average phenylalanine concentration measured in dried blood spots), attendance at follow-up appointments, and regular intake of protein substitutes without phenylalanine”.

  1. Comparison of PKU 1 and PKU 2 groups would be useful for all the results section.

Response: We added a paragraph at the end of the results section stating that “Summarizing the results of tables 1 2 and 3, PKU 1 and PKU 2 patients had similar BMI, body composition and bone mineral density. PKU 2 patients had higher Phe blood levels and lower Vitamin B 12 and D levels than PKU 1 patients. As expected PKU 2 patients had lower cholesterol Tyrosine, Vitamin B 12, Vitamin D and calcium intake than PKU 1 patients”.

Discussion:

  1. Would suggest re wording as not adequately compliant as they had no choice but to stop dietary treatment at 18 years.

Response: We changed the wording to “those who had to abandon the dietary treatment”.

  1. I would challenge your statement that it is only Phe concentrations leading to the differences in the findings there are multiple reasons for lower muscle health.

- Protein quality intake

- Vitamin and mineral intake this was different in the groups

- Higher phe may lead to less physical activity as decreased motivation to do exercise and this may affect muscle strength although this was not suggested in your findings.

- There were no markers on prealbumin which has been shown to affect muscle strength in PKU subjects and could be cited in your discussion.

- Other areas are BCAA intake which might have been a factor in those on a vegan diet

- The role of leucine in muscle function?

Response: We added a paragraph with the other factors that affect muscle and bone health suggested by the reviewer. Quote “However other factors explain the lower muscle and bone parameters in PKU 2 patients such as protein quality intake, lower vitamin and mineral intake and lower physical activity derived from the lack of motivation to have healthy lifestyles after having to abandon the dietary treatment. We did not measure prealbumin as an inflammatory marker influencing muscle function. Therefore, we cannot discard its pathogenic importance”.

  1. Similar study reference 33 need to expand on this and give a little more background. Not just PKU but need to be specific Firman concluded that regarding body protein status in PKU findings were inconclusive and the relationship between diet and protein status outcome is unclear. This was a detailed review and explored many factors that could influence protein status.

Response: We now state that “The author also concludes that new studies are required to confirm or refute the pathogenic importance of a lower protein intake on muscle function in PKU patients.

  1. Findings from 2 systematic reviews: BMD is it lower than the normal population reference range, but it is clinically normal, and this is an important finding. PKU patients on a diet have normal bone mineral density and are not at risk of increased fractures. I would quote the second reference on bone density in PKU.

Response: As suggested by the reviewer, we now state that “In our patients, BMD was lower than the normal population reference range, but it was clinically normal and we didn't find in their histories, report of fractures. Therefore, our PKU patients were not at a higher risk for fractures”.

  1. Mice are not the same as humans – this needs to be put into context and one cannot compare bone formation in mice and humans. Bone formation is a complex process and one animal model cannot be held up as an example of low bone density in PKU. This study looked at CGMP which is a different protein substitute type and leads to another possibility- again not proven - to benefit bone health.

Response: We now state that “although mice are not a good model to mimic bone formation in humans”.

  1. Main findings lower peak workload is an important observation but ideally need more explanation and the link with sarcopenia.

Response: We added a statement and a reference from our group about the association between peak workload and sarcopenia.

  1. Muscle mass is an important element for physical and functional ability and plays an underappreciated metabolic role in reducing risk of cardiometabolic diseases, obesity, cardiovascular disease, diabetes, and hyperlipidaemia- these points could be highlighted.

Response: We added “On top of its functional consequences, a lower muscle mass influences cardiovascular health, insulin sensitivity and thus glycemic status and serum lipid levels.”, to complement the statement.

  1. Limitations did not measure pre albumin which might give a better insight into protein status.

Response: We now state in the limitations section that we did not measure prealbumin.

  1. Muscle mass does not directly relate to functional ability which is a further limitation and needs more investigation.

Response: We also assessed muscle function measuring handgrip strength and walking speed.  These measurements evaluated functional ability in our patients.
